# Susceptibility to Secondary Bacterial Infections in Growing Rabbits Exposed to Ochratoxin A and Protected or Not by Herbal Supplements

**DOI:** 10.3390/toxins17100507

**Published:** 2025-10-15

**Authors:** Kalina Zhivkova, Stoycho Stoev, Vladimir Petrov, Vesselin Ivanov

**Affiliations:** 1Department of General and Clinical Pathology, Faculty of Veterinary Medicine, Trakia University, Students Campus, 6000 Stara Zagora, Bulgaria; 2Department of Veterinary Microbiology, Infectious and Parasitic Diseases, Faculty of Veterinary Medicine, Trakia University, Students Campus, 6000 Stara Zagora, Bulgaria; 3Department of Social Medicine, Health Management and Disaster Medicine, Faculty of Medicine, Trakia University, Students Campus, 6000 Stara Zagora, Bulgaria

**Keywords:** ochratoxin A, protective effect, herbal additives, *Silybum marianum*, *Withania somnifera*, *Centella asiatica*, silymarin, immunosuppression, pasteurellosis, rabbits

## Abstract

The protective effects of the herbal feed supplements *Silybum marianum*, Silymarin, *Withania somnifera,* and *Centella asiatica* against ochratoxin A (OTA) toxicity were studied in 48 New Zealand White rabbits (37-day-old) during an 80-day experiment. OTA was given at 2 ppm, whereas *Silybum marianum*, Silymarin, *Withania somnifera*, and *Centella asiatica* were given at feed levels of 5000 ppm, 25,000 ppm, 4000 ppm, and 4600 ppm, respectively. All rabbits were immunized against Rabbit Hemorrhagic Disease Virus (RHDV). OTA was found to induce an immunosuppressive effect on the humoral immune response. Reliable protection against OTA-provoked immunosuppression by Silimarin and *Withania somnifera* was found. The OTA-induced immunosuppression was responsible for secondary bacterial infection (pasteurellosis) and the death of two rabbits from the OTA-exposed group and one rabbit each from the groups protected with *Silybum marianum* and *Centella asiatica*. A decreased body weight was found in rabbits exposed to OTA, but the decrease was slighter in the rabbits protected with herbal supplements. The target organs damaged by OTA exposure were the liver, kidneys, and spleen, while weaker lesions were found in other internal organs, except in the cases of secondary pasteurellosis, in which the strongest damage was found in the lung. All investigated herbal supplements appeared to have stronger protective effects against OTA-induced damage to the kidneys and liver, with slightly protective effects observed in the lungs, myocardium, spleen, brain, intestine, testicles, and ovaries.

## 1. Introduction

Mycotoxins are global food-borne threats to animal health and are often involved as compromising factors in food safety [1]. Ochratoxin A (OTA) contamination of animal feed is frequently reported [2,3], and spontaneous cases of mycotoxicoses induced by OTA and some other mycotoxins are often reported in pigs and poultry [4,5]. OTA toxicity is attributed to its inhibitory effect on the enzymes involved in phenylalanine metabolism, the provoked increase in lipid peroxidation, the inhibitory effect on protein synthesis, and the suppression of mitochondrial phosphate transport and subsequent damage to mitochondria in the organs involved in its elimination or detoxification, e.g., the kidneys and liver.

Rabbits are reported to be very sensitive to OTA compared to other animals, such as guinea pigs, rats, and mice [6]. OTA was found to have a teratogenic effect [7] and to be extremely toxic for young rabbits, with an LD50 of 10 mg OTA/kg b.w. [8]. Unfortunately, there are only scarce data about spontaneous cases of ochratoxicosis in rabbits [9], which is in contrast to the fact that OTA is a common contaminant in rabbit feed. In a study in Argentina, 78% of all rabbit feed samples analyzed were reported to be positive for OTA [10]. In this regard, clinical symptoms and patholoanatomical, pathomorphological, immunosuppressive, and biochemical changes in ochratoxicosis in rabbits [11,12,13,14], as well as preventive measures [15], are poorly researched. Therefore, further studies in this area will be useful and will contribute to the diagnosis of this mycotoxicosis and the development of possible preventive actions.

A large number of plants or their extracts have been reported to be useful in alleviating some ailments or intoxication in animals or humans [16]. For example, more than 7500 plants or herbs in India have been used for medical treatment [17]. Traditional methods of treating diseases, intoxication, or ailments have been reported to contribute significantly to the discovery of some new drugs derived from herbs or plants [18]. Traditional medicine, such as medical plants or herbal extracts, is still used by the populations of developing countries to treat diseases and ailments. Unfortunately, only a small number of medicinal herbs and plants (about 15%) have been investigated for their biological activities and phytochemical compounds [19]. Recently, medicinal plants have been promoted as safe alternatives to synthetic drugs, and there has been increased interest in their use for medical purposes. Therefore, any knowledge related to their possible use for medical purposes, for example, knowledge obtained from animal experiments, could contribute greatly to the elucidation of their pharmaceutical properties and adequate application in the treatment of target ailments and diseases.

The use of medicinal plants as bioactive feed supplements to alleviate the toxic effects of OTA presents a good alternative to chemical protection, because such an approach is safe for animals and preserves the nutritional and organoleptic properties of the feed. The experiment conducted in this study aimed to clarify the protective effects of target herbal supplements, e.g., *Withania somnifera*, *Centella asiatica*, and *Silybum marianum*, against the toxic effect of OTA in rabbits. These herbs are known for their strong protective properties regarding the immune system, liver, and kidneys, which are usually compromised by OTA, and they also have strong antioxidant properties and could be useful against oxidative stress provoked by OTA [20].

*Silybum marianum* and its commercial seed extract Silymarin have been shown to have protective effects on the immune system, inflammation, oxidative stress, lipid peroxidation, cell membranes, liver, and kidneys [21,22,23,24,25,26,27,28]. A hepatoprotective effect of Silymarin was found in rats treated with the hepatotoxic substance NDEA (N-nitrosodiethylamine) [29]. A similar protective effect of *S. marianum* on the liver was reported in poults treated with aflatoxins [28], mice exposed to the hepatotoxic effects of alcohol [25], and rats exposed to diethylnitrosamine [23,24]. *S. marianum* was also reported to improve the health of growing rabbits [30]. *S. marianum* seeds have been found to decrease mortality/morbidity in rabbits with an improving effect on body weight and a possible protective effect against infectious diseases [31].

*Centella asiatica* is known for its protective and anti-inflammatory properties on the gastrointestinal mucosa and can improve the mucosal barrier against OTA-induced damage and the harmful effects of free radicals. The same herb can improve vascular integrity and reduce OTA-induced damage to the gastrointestinal system and vascular permeability [20,32]. *C. asiatica* has been reported to improve growth performance and feed conversion efficiency in rabbits [33], as well as the immune system and anti-bacterial defense [34,35], and it may alleviate OTA-induced immunosuppression and prevent possible secondary bacterial diseases [36,37,38]. The known protective effect of *C. asiatica* on the liver and kidneys in humans [34] or chickens [35] may alleviate OTA-induced damage in both organs in rabbits.

The third herb, *Withania somnifera*, whose possible protective effects have been investigated, shows similar anti-inflammatory, immunostimulatory, hepatoprotective, and antioxidant effects in some laboratory animals [39,40,41] and chickens [35], as well as the ability to reduce oxidative stress and improve health status in rabbits [42].

OTA is a mycotoxin produced mainly by *Aspergillus occhraceus* [43] and is involved in mycotoxic nephropathy in Balkan countries [44]. The aim of the present experiment was to clarify the possible protective effects of the herbal supplements *C. asiatica*, *W. somnifera*, *S. marianum*, and Silymarin applied to the rabbit diet against the toxic effects of OTA. The same natural bioactive supplements to rabbit feed will be evaluated for possible application in real practice in order to ensure the safe utilization of feed naturally contaminated with OTA, avoiding possible growth losses and secondary diseases in stock rabbits, as well as possible scrapping of OTA-contaminated feeds.

## 2. Results

### 2.1. Clinical Signs, Feed Intake, and Clinical Microbiology

Reduced feed intake, depression, lethargy, and single cases of anorexia were seen only in rabbits treated with 2 ppm OTA in the diet and not protected with herbal supplements. These signs were observed mainly in the last weeks of the study. In addition, four rabbits from the OTA group (50%) and a few rabbits from the groups supplemented with *C. asiatica* (CA + OTA group) (two rabbits–25%), *S. marianum* (SM + OTA group) (one rabbit–12.5%), and *W. somnifera* (WS + OTA group) (one rabbit–12.5%) had difficulty breathing and a runny nose or snuffles observed at the end of the second week of the experiment, and some of them died within a few days, as follows: two rabbits (one male and one female–25%) from the OTA group died on day 12 of the experiment, one female rabbit from the CA + OTA group (12.5%) died on day 13, and one female rabbit from the SM + OTA group (12.5%) died on day 14 from the beginning of experiment. Subsequent microbiological investigation of the dead rabbits revealed an abundant presence of *Pasteurella cuniculiseptica* in the lung. In order to protect against possible secondary bacterial infections, all rabbits were treated with Baytril at that time (from day 14 to day 19 of the experiment).

### 2.2. Body Weight and Feed Intake

A significant decrease in body weight was observed on day 80 of the experiment in rabbits exposed only to OTA, but that decrease was slighter in the groups protected with herbal supplements. The slightest decrease in body weight compared to the control group was seen in the rabbits protected with *C. asiatica* (Table 1). Feed intake was decreased in rabbits from the OTA-exposed groups, but that decrease was less pronounced in the groups protected with herbal supplements.

### 2.3. Immunological Findings

The measurement of the hemagglutination-inhibiting antibody titer (HIAT) on day 28 after immunization against Rabbit Hemorrhagic Disease Virus (RHDV) revealed that only the control group and the groups protected with Silymarin and *W. somnifera* against OTA-toxicity developed reliable protection against RHDV infection. A slight increase in HIAT was also observed in the group protected with *S. marianum*, but the same protection was not sufficient to prevent development of RHDV infection. The HIAT was lowest in the group treated only with OTA, followed by the group protected with *C. asiatica*, in which no protection against RHDV infection was observed (Table 2). Obviously, a significant and reliable protective effect against OTA-induced suppression of the humoral immune response was found only in rabbits supplemented with *W. somnifera* and/or Silymarin, while the protective effect of *S. marianum* supplementation was slighter and insufficient to prevent such disease.

### 2.4. Macroscopic Changes at Necropsy

The macroscopic changes at necropsy revealed slight signs of weight loss only in the rabbits of the OTA group not protected with herbal supplements. Mild hyperemia of the mucosal surface of intestine was found, and it was more obvious in the rabbits of the OTA group not protected with herbal supplements. The liver of the rabbits in the OTA group showed a mottled surface with pale grey-brown staining in some areas of the surface (Figure 1A). The kidneys of rabbits from the same group showed a paler or mottled color of the cortical part, well visible on the cut surface (Figure 1B). A striated surface and mild fibrosis of the kidneys were observed only in rabbits from the same group euthanized on the 80th day of the experiment (Figure 1C). A small amount of mucus in the intestinal content was seen mainly in the rabbits of the OTA-exposed group. A faintly mottled surface of the lung and hyperemia or mucous fluid on the mucosal surface of the nasal cavity and trachea were found in a few rabbits in the OTA group. Similar, but less pronounced, macroscopic changes in the liver, kidneys, and lungs were also observed in a few rabbits from the groups protected with herbal supplements.

In two rabbits from the OTA group and in one rabbit from the group protected with *S. marianum* (SM + OTA group), which were euthanized on the 80th day of the experiment, some changes localized mainly in the nasal cavity and trachea or lung were observed. The nasal mucosa was hyperemic with small erosions and ulcers on the mucosal surface. Purulent yellow-green fluid (purulent matter) was also found on the nasal or tracheal mucosa. Small dried crusts of secretions around the nose were also seen. A mottled surface of the lung was seen in the same rabbits (Figure 1D). The liver and kidneys of the same rabbits had a pale or mottled color and a friable consistency, probably as a result of the advanced parenchymatous degeneration.

In rabbits that died of pasteurellosis, hemorrhages were observed in the lungs and rarely on the liver surface. The trachea and bronchi contained mucopurulent fluid. The lungs were hyperemic and edematous. Catarrhal or purulent bronchopneumonia, as well as croupous pneumonia was found in the lung tissue, including a mottled surface of the lung and an enlargement of the pulmonary interstitium. Adhesion between the two layers of the pleura or pericardium (pleurisy and pericarditis) was also seen as a common pathological feature (Figure 1E). Small yellow or grey-yellow foci (abscesses) were found in the lungs (Figure 1F), which were scattered mainly in the apical lobes. Serous or hemorrhagic exudate was found in the thoracic cavity. An increased amount of fluid was also observed in the abdominal cavity.

### 2.5. Pathomorphological Changes

#### 2.5.1. Pathomorphological Changes on the 30th and 80th Day of the Study

Microscopic investigation of the internal organs of rabbits on the 30th and 80th days showed stronger pathological changes in rabbits from the OTA group, which was not protected with herbal supplements, followed by the other experimental groups (Table 3).

Histological investigation of the liver revealed granular and rarely vacuolar degeneration in hepatocytes, in addition to karyopyknosis and karyolysis of hepatocytes, activation of capillary endothelium and Kupffer cells (Figure 2A), edema or fibrosis around bile ducts, perivascular infiltration of mononuclear cells, bile duct proliferation, and mild vascular congestion. These lesions were better expressed in rabbits in the OTA group compared to the groups protected with herbal supplements (Table 3). In addition, the aforementioned degenerative and especially proliferative lesions were better pronounced on the 80th day of the study in rabbits in the OTA group (Figure 2B).

Histological examination of the kidneys revealed severe granular degeneration in the proximal tubular epithelium in rabbits in the OTA group. Disintegration and desquamation of the proximal tubular epithelium (Figure 2C) and hyaline or cellular casts in the tubular lumen were also found in the same rabbits, in addition to interstitial mononuclear proliferation and activation of the capillary endothelium. Vascular or capillary congestion and mild hypercellularity (endothelial or mesanglial proliferation) in the glomeruli were also seen. The lesions described above, accompanied by connective tissue proliferation in the interstitium or glomerulesar sclerosis and tubular atrophy, were well manifested on the 80th day from the start of the experiment (Figure 2D). The same changes were better expressed in rabbits from the OTA group without any herbal protection and were less pronounced in the groups protected with herbal supplements (Table 3). Mild connective tissue proliferation and tubular atrophy were mainly observed in rabbits protected with *W. somnifera* but rarely seen in rabbits supplemented with other herbal protectors on the 80th day of the study (Figure 2E). Glomerular sclerosis was found only in rabbits from the OTA + SM, OTA + WS, and OTA + CA groups at this time. Mild local connective tissue proliferation was rarely found in single rabbits in the OTA + SIL group on the 80th day of the study (Table 3).

Microscopic examination of the lung revealed mild vascular congestion, siderocytes in the peritubular capillaries, activation of alveolar macrophages, focal catarrhal pneumonia, and proliferation of bronchial or alveolar epithelial cells, as well as mononuclear peribronchial or perivascular proliferation, in rabbits in the OTA group (Figure 2F), but these lesions were less pronounced in the groups protected with herbal supplements. The same lesions were of the same intensity or more pronounced on the 80th day of the study, especially in rabbits in the OTA group that were not protected with herbal supplements (Table 3).

Histological investigation of the heart revealed mild degenerative and lytic lesions or irregular staining of some myofibrils (increased eosinophilia) (Figure 2G), mild vascular congestion, and mononuclear infiltration, which were seen mostly in rabbits from the OTA group that were not protected by herbal supplements, but the same changes were mild and rarely seen in the groups receiving herbal protectors. The same lesions with similar intensity were seen on the 80th day of the study (Table 3).

Microscopic examination of the spleen revealed a reduction in white pulp, in addition to degenerative lesions and/or cellular depletion in the lymphoid tissue, with red pulp showing a significant predominance over white pulp (Figure 2H). Lymphoid cell depletion was also observed in the cortical part of the thymus. These lesions were most pronounced in rabbits from the OTA group, followed by the OTA + CA and OTA + SM groups, but were rarely found in rabbits from the OTA + SIL and OTA + WS groups. No differences in the intensity of these lesions were observed on the 30th and 80th days of the study (Table 3).

Histological investigation of the intestine revealed degenerative lesions in the superficial and glandular epithelium of the mucosa and mild vascular congestion, observed mostly in rabbits in the OTA group (Figure 3A), and these lesions were less pronounced in rabbits protected with herbal supplements. No differences in the intensity of the microscopic findings were observed between the 30th and 80th days of the study.

Histological examination of the brain revealed lytic lesions in the brain substance (Figure 3B), lysis, or pykosis in the tigroid substance in neurons (tigrolysis or tigropyknosis) (Figure 3C), which were mostly observed in rabbits in the OTA group, but the same lesions were not so obvious or were rarely seen in rabbits protected with herbal supplements. Focal proliferation of microglia around the damaged neurons and pericapillary infiltration with lymphocytes was found only in rabbits in the OTA group not protected with herbal supplements and was most pronounced on the 80th day of the study (Table 3).

Microscopical investigation of the cerebellum revealed edematous and lytic lesions in the region of Purkinje cells (Figure 3D) as well as in the *lamina medullaris,* mostly found in rabbits in the OTA group and rarely in rabbits receiving herbal protectors. The same lesions were observed on the 30th and 80th days of the study (Table 3).

Histological investigation of the ovary revealed mild congestion, connective tissue proliferation, and mild interstitial oedema, seen mostly in rabbits in the OTA group on the 30th day or 80th day of the study (Figure 3E). Similar changes, but less pronounced, were occasionally found in rabbits from groups supplemented with herbal protectors (Table 3).

Microscopical examination of the testicles revealed mild degenerative lesions in the seminiferous tubules (Figure 3F) and/or mild edema in the interstitium, mainly in rabbits in the OTA group on the 80th day of the study, but such lesions with lower intensity were rarely seen in groups supplemented by herbal protectors (Table 3).

Histopathological examination of the internal organs of control rabbits did not reveal any pathological lesions.

#### 2.5.2. Pathomorphological Changes in Rabbits That Died of Pasteurellosis

Histological examination of the lung in rabbits that died of pasteurellosis revealed purulent pneumonia with accumulation of leucocytes and small abscesses in the lungs. In other areas of the lung, the characteristic lesions of croupous pneumonia were found with accumulation of fibrin in the alveoli and interstitial tissue (Figure 3G) and thrombosis (fibrin clots) in the vessels. In the liver, severe focal degeneration or necrotic changes in hepatocytes and accumulation of fibrin in the vessels were seen (Figure 3H). In the kidneys, thrombosis (fibrin clots) in the vessels was found, in addition to damage in the proximal tubules characteristic for OTA intoxication. In the heart, severe degenerative and lytic changes in some myofibrils were found. In the spleen, a reduction in white pulp and a predomination of red pulp was seen. Histological lesions in the brain, cerebellum, intestine, testicles, and ovaries showed no other changes, except for the lesions already described above characteristic for the OTA group on the 30th day of the study.

## 3. Discussion

After analyzing the clinical signs, macroscopic and histological lesions in internal organs, body weight changes, and immunosuppression of the humoral immune response (decrease in antibody titer against RHDV), it can be concluded that these changes were most pronounced in rabbits from the OTA group without herbal protection. Regarding the herbal protectors used in this study, it can be seen that all the herbs studied have a good protective effect against kidney or liver damage, as found by histological examination, while Silymarin and *W. somnifera* showed the best protective capabilities against the immunosuppressive effect of OTA (given at a feed level of 2 ppm) on the humoral immune response (HIAT). It was found that the strong immunosuppressive effect of OTA on the antibody titer against RHDV was completely eliminated in rabbits treated with the same herbal supplements (Silymarin and *W. somnifera*) and only slight protection was seen in rabbits treated with *S. marianum*, but no protection was observed in rabbits treated with *C. asiatica.* This immunological finding was further confirmed by histopathological changes in the spleen or thymus, where degenerative changes in and depletion of immunocompetent cells were weaker or almost absent in the same experimental groups, confirming the good immunostimulatory and/or immunoprotective properties of Silymarin and *W. somnifera*. Surprisingly, a stronger humoral immune response was even found in rabbits fed an OTA-exposed diet and protected by Silymarin compared to control rabbits fed an OTA-free diet. Due to ethical considerations and limitations in the number of animals used, as well as the dose–response limitation (only one feed level of OTA was studied) and gender differences inside the groups, the observed statistical fluctuations between groups were not always sufficient to establish confidential differences between them.

The reported increase in blood urea nitrogen (BUN) and serum creatinine in the OTA-treated rabbits [13,45] could explain the strong pathological changes in the kidneys found in the present study as well as by some other authors [46,47].

The decreased levels of red blood cell (RBC) count and hemoglobin in the OTA-exposed rabbits [48,49] may be the cause of increased anaerobic metabolism and subsequent impairment of oxygen supply to tissues and organs, which could explain the degenerative lesions observed in internal organs.

The observed degenerative changes in the liver of the OTA-exposed rabbits could be responsible for the reported changes in lipid metabolism, including increased triglycerides and cholesterol in such rabbits [50]. On the other hand, enhanced lipid peroxidation, which is reported as a consequence of OTA exposure [37,51], may provoke further damage to cellular membranes and influx of cellular calcium and subsequent cell necrosis [52].

The reported increase in serum levels of ALT, AST, and ALP in OTA-exposed rabbits [13,45] may explain the observed liver lesions in such rabbits found in the present study.

The low body weight gain in the rabbits exposed to OTA may be a consequence of OTA-induced protein synthesis disorders [35,36,53,54,55], as well as some lesions in the gastrointestinal system, as found in this experiment, and subsequent disturbances in nutrient absorption [56].

The macroscopic changes in the liver and kidneys (pale or mottled surface) and the observed lesions in the parenchyma in both organs could be due to the elimination pathway of OTA through the kidneys and liver, as well as to the enterohepatic recirculation of OTA [57,58], which has a direct toxic effect on the same organs [55]. The accumulation of OTA in the proximal tubular epithelium has been reported to be related to the membrane organic anion transport pathway of OTA elimination through the same tubules [59]. The high OTA content reported in the liver, kidneys, and myocardium of rabbits exposed to 1 ppm or 2 ppm OTA in the diet [11,12] may explain the severe macroscopic and histological lesions seen in these organs in this study or in other similar studies [45,47]. Residual amounts of OTA have been reported in the same organs even 4 weeks after rabbits were stopped from being fed the OTA-contaminated diet [11,12], which may explain why the most severe lesions were found in these organs.

Similar histological lesions in myofibrils of myocardium, including lytic changes and mononuclear proliferation, observed in the present study have been also reported in the same breed by other authors [45] or in newborn rabbits receiving OTA through their mothers [47].

The high mortality observed in rabbits fed an OTA-contaminated diet [60] appears to be a consequence of secondary bacterial infection, mainly provoked by *Pasteurella cuniculiseptica (multocida)* as a result of the immunosuppressive effect of OTA observed in many animal species [36,37]. Pasteurellosis in rabbits is an endogenous infection of a sporadic or enzootic nature, and no pronounced contagiousness has been established. *Pasteurella cuniculiseptica* is a facultative pathogenic bacterium that saprophytes on the mucosa of the upper respiratory tract of rabbits. Due to the fact that rabbits are normal carriers of the bacilli, there is a close relationship between the resistance of the organism and the virulence of the causative agent, which explains the very diverse forms of this disease, such as runny nose, purulent or croupous pneumonia, etc. The disease occurs as a result of predisposing factors such as stress or immunosuppression, which were provoked by OTA exposure in the present experiment. The body’s resistance is of particular importance, but the spread of infection can also occur through the respiratory system via the sprays formed during sneezing and coughing, which explains the damage to the nasal cavity, trachea, and pneumonic changes in the lungs, seen mainly in rabbits from the OTA group not protected with herbal supplements. In the chronic form, the changes are localized mainly in the nasal cavity [61].

The reduction in white pulp and degenerative lesions, as well as cellular depletion of lymphoid tissue in the spleen and thymus in the OTA-treated rabbits [62], has been associated with impaired protein synthesis in lymphocytes, which is responsible for the impairment of their proliferation and differentiation [35,63,64]. The observed decrease in the number of immunocompetent cells in the OTA-compromised rabbits is likely the reason for the observed suppression of the humoral immune response. The immunosuppressive effect of OTA has been previously reported in various animals or poultry exposed to OTA in some previous experimental investigations [36,37,65]. Such suppression was also reported as the first expressed toxic effect of OTA, contributing to the occurrence of secondary microbial infections in OTA-compromised animals [37].

A study on the possible interaction between OTA and *Pasteurella multocida* in rabbits revealed that OTA could significantly worsen *P. multocida* infection, contributing to increased mortality in rabbits [66], which could be explained by the decrease in humoral and cellular immune responses provoked by OTA in rabbits [46], as well as the decrease in lymphoid cells in the spleen, mesenteric lymph nodes, and thymus in such rabbits [12]. The deterioration of the general condition of the rabbits that were intranasally infected with *P. multocida* and additionally exposed to OTA was demonstrated by the increase in the number of microorganisms in these rabbits when compared to rabbits infected only with *P. multocida* [62].

The beneficial effect of Silymarin or *S. marianum* on body weight found in this experiment has also been reported in aflatoxin-compromised chickens [67]. It was established that feed intake, feed conversion ratio, and body weight gain were improved in chickens given the same herbal supplements to their diet, and these protective effects were similar to the protective effect of a toxin binder [28]. A similar beneficial effect on body weight has also been reported in rats given *C. asiatica* [68] or *W. somnifera* [40] via their diet. These studies support our experimental investigations and the observed increase in body weight gain in rabbits given the same herbal supplements along with OTA, with the most potent protection being observed with *C. asiatica*.

Silymarin has also been reported to be a potent kidney and liver protector against the toxic effect of salinomycin in rabbits [69]. Silymarin has also been shown to be a potent protector against liver lesions in rabbits provoked by isoniazid [70], as well as against hematological and biochemical changes in rabbits provoked by nickel chloride [71]. A similar protective effect of Silymarin has also been reported against gentamicin-induced renal lesions in dogs [72] and alloxan-induced diabetic nephropathy in rats [73]. A similar protective effect against hepatocellular damage in rats induced by N-nitrosodiethylamine (NDEA) and carbon tetrachloride (CCl4) [29] or in chickens induced by aflatoxins [28,67] has also been reported for *S. marianum,* as evidenced by decreased serum levels of ALT, AST, and ALP, which are characteristic biomarkers for hepatocellular damage. The hepatoprotective and nephroprotective effects of Silymarin and *S. marianum* were demonstrated in this study, as observed by less pronounced histological lesions in the liver and kidneys of rabbits supplemented with the same supplements along with OTA exposure.

The intimate mechanism of the protective effect of the herb *S. marianum* and the commercial seed extract of the same herb Silymarin is due to the reduction in lipid peroxidation and the increase in endogenous antioxidants, ensuring the integrity of cell membranes [23,74]. This protection is mainly due to the target flavonoids, including silybin, silychristin, isosilybin, and silydianin [22]. Among them, silybin is considered the strongest protector against various toxic substances with a powerful protection effect on the kidneys and liver [22,75]. *S. marianum* has also been found to be a good protector against target mycotoxins such as OTA and aflatoxins in chicken feed [28,35,67].

The other herb, *W. somnifera,* has also been reported to have potent liver protection capability against carbon-tetrachloride-induced liver lesions in rats and to be a potent antioxidant [41], which was demonstrated in the present study. The intimate mechanism of protection of this herb is attributed to biologically active alkaloids such as isopelletierine and anaferine, steroidal lactones such as withanolides and withaferins, and some saponins [40].

The potent protective properties of *W. somnifera*, *S. marianum*, or Silymarin against OTA-induced immunosuppression on humoral immunity in the present experiment are consistent with other reports regarding immunostimulatory effect of these herbs on humoral or cellular immunity [21,39].

The protective effects of *C. asiatica* have also been reported in previous experimental investigations, including protection of the gastro-intestinal system and vessels [20,32,35]. A similar protective effect was seen in the present experiment against OTA-induced pathological lesions in the intestinal mucosa. Similar protective effects of *C. asiatica* have also been found on the kidneys and liver [34,35], and the same protective properties were demonstrated in the present experiment. However, the mild-to-moderate immunostimulatory properties of *C. asiatica,* as previously suggested by Oyedeji and Afolayan [34] or Stoev et al. [35], were not seen in this experimental study in rabbits.

## 4. Conclusions

Susceptibility to a natural endogenous infection (pasteurellosis) has been demonstrated for the first time in rabbits exposed to the immunotoxic effect of OTA, which serves as a predisposing factor. The OTA-induced suppression of humoral immunity, defined in principle, is demonstrated in practice by the present experiment. Immunosuppression appears to be the first pronounced toxic effect of OTA that may manifest clinically before nephropathy. Humoral immunity was affected to a degree that allowed for the development of clinical disease and death in rabbits at only 2 ppm OTA in the diet. The extent to which OTA may interact with other mold metabolites in commercial rabbit feeds, such as penicillic acid, which may be synergistic with OTA in immunosuppression, may also influence the significance of the relatively lower doses of OTA that commercial rabbits may encounter in some feeds. The present study may explain both the high mortality rate in commercial rabbits and the complex clinicomorphological picture of ochratoxicosis in rabbits presented by some authors, in which the immunosuppressive dimension and secondary pasteurellosis are not recognized.

The target organs damaged by OTA exposure in rabbits were found to be the liver and kidneys, as well as the immune system (including the spleen and thymus), with the most severe lesions found in these organs. Less severe lesions were found in the heart, intestine, and lung, except in cases of secondary pasteurellosis, when the most severe damage was found in the lungs.

It seems that all the herbal supplements or herbal products (Silymarin) tested in the present study could be applied “in addition” to target commercial mycotoxin binding agents to alleviate the toxic effects of OTA-contaminated rabbit feed. These herbal supplements, especially Silymarin and *W. somnifera*, could be used as a practical approach for the safe utilization of OTA-contaminated feed for rabbits, preventing or at least alleviating the harmful effects of OTA on health, the immune system, and body weight [35,38,76] and reducing potential economic losses from reduced rabbit weight, rejection of such mycotoxin-contaminated feed, or possible secondary bacterial infection.

## 5. Materials and Methods

### 5.1. OTA Supply

The supply with OTA was ensured by using *Aspergillus ochraceus* strain (isolate D2306), as described previously by Stoev et al. [37,43]. Mycotoxin production was performed on sterilized and moistened shredded wheat (40 g) in 500 mL conical flasks, moisturized by a 40% (*v*/*w*) addition of sterile water and incubated on a rotary shaker at 27 °C for 14 days [43] and then sterilized at 80 °C for 1 h and stored at −20 °C until usage. HPLC analysis of a sample of the shredded wheat was found to be contaminated with ~2 mg/g OTA and a very small amount of the biologically inactive dechloro-analogue ochratoxin B. This detection method is appropriate for analysis of material of a high OTA concentration and for monitoring other mycotoxins. No other mycotoxins were found, and the subsequent dilution by nearly 10^3^ when homogenized into the rabbit diet ensured only a minimal content of any other components. The OTA-rich material was subsequently homogenized into the rabbit feed in such a way as to provide the necessary level of 2 ppm OTA in the diet. The final OTA content in the feed was also studied by HPLC, as previously described [44] in other studies to confirm the required level of 2 ppm OTA.

A multi-mycotoxin extraction method (multi-mycotoxin screen) was used to analyze the feed samples [44]. In the same method, two extracts could be generated in one step: a neutral fraction and an acid fraction (containing OTA and some other mycotoxins).

A two-dimensional thin-layer chromatographic (TLC) technique was used initially, as 20 µL of the acid fraction (containing OTA) dissolved in dichloromethane (DCM) was spotted on TLC plates and subsequently dried in a warm stream of air. The spotted plates were then developed in TLC tanks using two solvents, Diethyl Phthalate (DEP) and Toluene–Ethyl Acetate–Formic Acid (TEF) in two-dimensional directions and were dried after each development. The plates were visualized under ultraviolet (UV) light at 366 nm for the presence of fluorescence and compared to the OTA standard plate. The blue fluorescent spot with retardation factors RF1 (50) and RF2 (30) for OTA was then compared with the RF values of the OTA standard [44].

HPLC analysis and quantification of OTA was performed using a Shimadzu system (Kyoto, Japan) with a fluorescence detector (RF-10AXL) and a Symmetry column (250 × 4.6 mm internal diameter), as previously described [44]. The procedure included the following parameters: excitation and emission wavelength—333 and 477 nm, mobile phase—acetonitrile–water (50:50, *v*/*v*), flow rate—0.8 mL/min, and injection volume of analyte and standard—10 μL per sample [44]. The limit of detection (LOD) was about 0.03 µg/kg (ppb), and the average recovery was about 88%. The peak areas and retention times of OTA were used to determine the amount of mycotoxin per sample based on that of the standard.

The standard feed, commercially available for rabbit farming, did not contain any additional mycotoxins.

### 5.2. Herbal Supplements

Herbal supplements (*S. marianum*, *C. asiatica*, *W. somnifera*) were purchased from Parceval Pharmaceuticals (Wellington 7654, South Africa) and consisted of *S. marianum* fruits powder (400 μm particles), *W. somnifera* radix powder (400 μm particles), and *C. asiatica* leaf powder (400 μm particles), provided in vacuum packs. Silymarin, which is a powdered extract of milk thistle, was manufactured by Wuxi Gorunjie Technology Co., LTD, China. The doses of herbal supplements were calculated by considering the dose conversion factors available in the literature and the body weight of the rabbits, the WHO norms, and other toxicological guidelines, as described previously [38], to be close to the effective doses against OTA toxicity. Available data from experimental studies were also taken into consideration [20,28,35,41]. Some additional adjustments were made based on the rabbit breed and age of the rabbits.

### 5.3. Study Design

The experiment was conducted with 48 rabbits of the New Zealand White breed. The rabbits were 30 days old and weighed 0.600–0.700 kg, and they were purchased from Agricultural Institute “NIGO” Stara Zagora. They were housed in the animal breeding complex of the Agricultural Faculty of Trakia University and were raised in the standardized conditions required for this breed of rabbits, with a natural temperature, humidity, and light regime appropriate for their age. Standard complete feed for growing rabbits and drinking water were available *ad libitum*. The rabbit housing facilities met the requirements for a floor area of 3500 cm^2^ per rabbit and a minimum height of over 45 cm. Within one week of their purchase (adaptation period), the rabbits were fed with a standard complete feed for growing rabbits, free from mycotoxins, and containing all necessary mineral supplements and vitamins. The feed was purchased from Grainstore feeds AD, Sofia, Bulgaria, and had the following structure and composition:

Feed composition: alfalfa meal, barley, oats, wheat bran, sunflower meal, soybean meal, wheat, corn, sugar beet granules and some minerals, incl. calcium carbonate, monocalcium phosphate, L lysine, DL methionine, sodium chloride, lecithin, vitamin-microelement premix, L threonine, sodium bicarbonate, rice bran, choline chloride, sunflower oil, and acidifier.

Analytical composition of feed: crude protein (17%), crude fat (3.6%), crude fiber (15%), crude ash (7.3%), calcium (0.91%), phosphorus (0.52%), and sodium (0.27%).

Additives in 1 kg of feed: vitamin A—6000 IU/kg, vitamin D3—600 IU/kg, vitamin E—24.58 IU/kg; trace elements (mg/kg): ferrous sulfate monohydrate (106.6), copper sulfate pentahydrate (18.5), manganese oxide (72.6), zinc oxide (96.5), calcium iodate (1.06), selenium (0.45), and antioxidants (mg/kg), including E321 (4.0), E320 (1.5), and E330 (2.0).

The rabbits were randomly assigned to one control and 5 experimental groups consisting of 4 female and 4 male rabbits in each group.

Standard complete feed for growing rabbits (non-pelleted) was mixed with OTA and herbal supplements as follows: OTA group—receiving standard complete feed containing 2 ppm OTA; WS + OTA group—receiving standard complete feed containing 2 ppm OTA and 4000 ppm *W. somnifera*; CA + OTA group—receiving standard complete feed containing 2 ppm OTA and 4600 ppm *C. asiatica*; SM + OTA group—receiving standard complete feed containing 2 ppm OTA and 5000 ppm *S. marianum*; SIL + OTA group—receiving standard complete feed containing 2 ppm OTA and 25,000 ppm Silymarin (or 2.5% of the feed); control group—receiving standard complete feed without any mycotoxins or herbal supplements (Table 4). The herbal supplements and OTA were carefully homogenized in the standard ration by stepwise mixing with the standard complete feed intended for each group. The same feeds were then pelleted using a special electric feed pelletizer GF-0893. After each feed change, careful cleaning and disinfection were carried out to prevent any possible mixing of feed intended for different groups. The thus-prepared feeds (containing or not OTA and herbal supplements) and drinking water were available *ad libitum*, and the same were added daily for 80 days.

The following vitamins, coccidiostats, and minerals were given to the rabbits according to standard prophylactic programs for growing rabbits and according to the respective prescriptions and age of the rabbits: egg booster at 1 g/L H_2_O (from day 1 to 4 and from day 14 to 20 after purchasing rabbits), EsB3 at 1 g/L H_2_O (4 days within the adaptation period), and Baytril (KVP Pharma, D-24106 Kiel, Germany) at 5 mg/kg i.m. (from day 14 to 19) in order to protect against pasteurellosis.

### 5.4. Immunization

All rabbits (control and experimental) were immunized at 37 days of age (immediately after the start of the experiment) against RHDV (with commercial vaccine Pestorin, manufactured by Bioveta, a.s., Komenskeho 212, 683 23 Ivanovice na Hane, Czech Republic) at a dose of 1 mL subcutaneously to each animal.

### 5.5. Serological Examinations and Assessment of Immune Response

Blood for serological examination was collected on day 1 (immediately before immunization) and on day 28 after immunization. A total of 1 mL of blood was taken from the *v. auricularis* (or *v. jugularis*) of each rabbit, after prior sedation with *Acepromazine* (The Netherlands, 3440 AB Woerden) at a dose of 0.75 mg/kg i.m. The blood was centrifuged within 1 h of collection, after which 100 µL of serum was separated from each sample for serological examination.

The humoral immune response was examined by measuring the antibody titer for Rabbit Hemorrhagic Disease (RHD) using INgezim Rabbit R.17.RHD.K1 (Spain, EUROFINS, vaccine batch number 140191_01), based on an indirect ELISA technique for detection of specific antibodies to Rabbit Hemorrhagic Disease Virus (RHDV), which uses peroxidase-conjugated protein A and a recombinant antigen (VP1 protein of RHDV). Before analysis, serum samples were diluted 1/200 in the diluent provided (5 μL serum sample in 995 μL diluent). A total of 100 µL of the diluted samples (positive and negative control serum) was added to the wells of the plate, and the plate was sealed and incubated for 1 h at 37 °C. It was washed 3 times, and peroxidase conjugate (protein A) was added to each well, sealed again, and incubated for 1 h at room temperature (20–25 °C). If the serum sample contained antibodies specific for RHDV, the conjugate bound to them, while if it did not contain specific antibodies, no binding was detected. It was washed again 5 times, and 100 µL of the substrate solution was added to each well, which in the presence of the peroxidase developed a colorimetric reaction. The plate was stored again at room temperature for 10 min. Subsequently, 100 µL of stop solution was added to each well, following the same order in which the substrate was added. Optical density (OD) values were read with a spectrophotometer at 450 nm within 5 min after addition of the stop solution.

Assay validation was performed when “OD Positive control/OD Negative control > 10”. Cut off calculation: cut off positive/negative was 0.300. Samples with an OD higher than 0.300 were considered positives, and samples with an OD lower than 0.300 were considered negatives. The sera titer was the highest dilution with an OD higher than 0.300.

When reporting the results, it was taken into account that rabbit samples with an optical density value (at a dilution of 1/200) above 0.9 indicated sufficiently developed immune protection against RHDV infection.

### 5.6. Clinical Microbiology

Lung and liver samples from rabbits that died on day 12, 13, and 14 of the study were subjected to standard microbiological investigation in the department of Veterinary Microbiology, Infectious and Parasitic Diseases, Faculty of Veterinary Medicine, Trakia University.

### 5.7. Body Weight and Weight of Feed Consumed

The body weight of the rabbits was measured on the first day of the experiment—b.w. ranged between 0.800 and 0.900 kg. Also, body weight and feed intake were monitored daily throughout the experiment.

### 5.8. Pathomorphological Investigations

Four rabbits (two males and two females) from all experimental groups were euthanized on day 30 from the beginning of the experiment. Euthanasia was performed by an overdose of Isoflurane (by inhalation) and pre-treatment with a sedative Zoletil—i.m. (Virbac, Carros, France). The remaining animals from each group were euthanized on day 80 after the beginning of experiment in the same way. Materials for pathomorphological investigations of the euthanized rabbits were taken from the kidneys, liver, lungs, heart, intestines, brain, cerebellum, testicles or ovaries, spleen, thymus, and lymph nodes. The same materials were fixed in 10% neutral buffered formalin or processed for freezing microtome. The fixed materials were subsequently embedded in paraffin, sectioned at 6 μm, and stained with hematoxylin–eosin. Some of the embedded tissues were also stained with periodic acid–Schiff (PAS) to detect glycoproteins, mucoproteins, or lipoproteins in the tissues and cells. Some of the embedded materials were also stained with Weigert iron hematoxylin to detect the presence or absence of fibrin. Frozen materials were stained by Sudan III for detecting fat.

### 5.9. Statistical Methods

A one-way ANOVA and Tukey’s test as a post hoc test were used to assess significant differences between the mean values of the studied parameters. This was achieved using the statistical software GraphPad InStat version 3.10 (San Diego, CA, USA). The level of significant difference was set at *p* < 0.05.

## Figures and Tables

**Figure 1 toxins-17-00507-f001:**
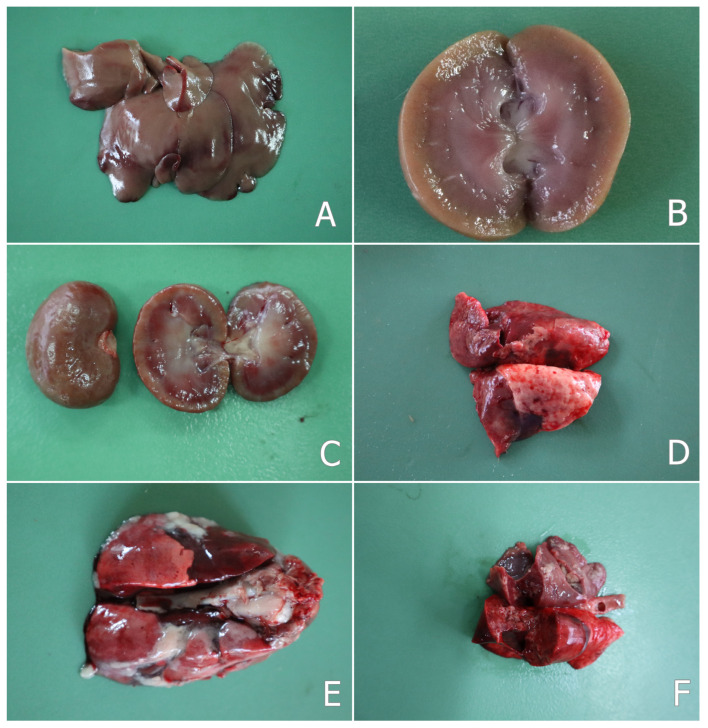
(**A**) Mottled surface of liver in a rabbit from the OTA group on the 30th day of the study; (**B**) pale cortex of kidney—cut surface of a kidney in a rabbit from the OTA group on the 30th day of the study; (**C**) furrowed surface of kidney—surface and cut surface of a kidney in a rabbit from the OTA group on the 80th day of the study; (**D**) mottled lung surface in a rabbit from the OTA group on the 80th day of the study; (**E**) sero-fibrinous pleurisy and pericarditis, and adhesion between the two layers of the pleura in a rabbit from the OTA group that died of pasteurellosis on the 12th day of the study; (**F**) purulent pneumonia and small yellow or grey-yellow foci (abscesses) in the lungs in a rabbit from the OTA group that died of pasteurellosis on the 12th day of the study.

**Figure 2 toxins-17-00507-f002:**
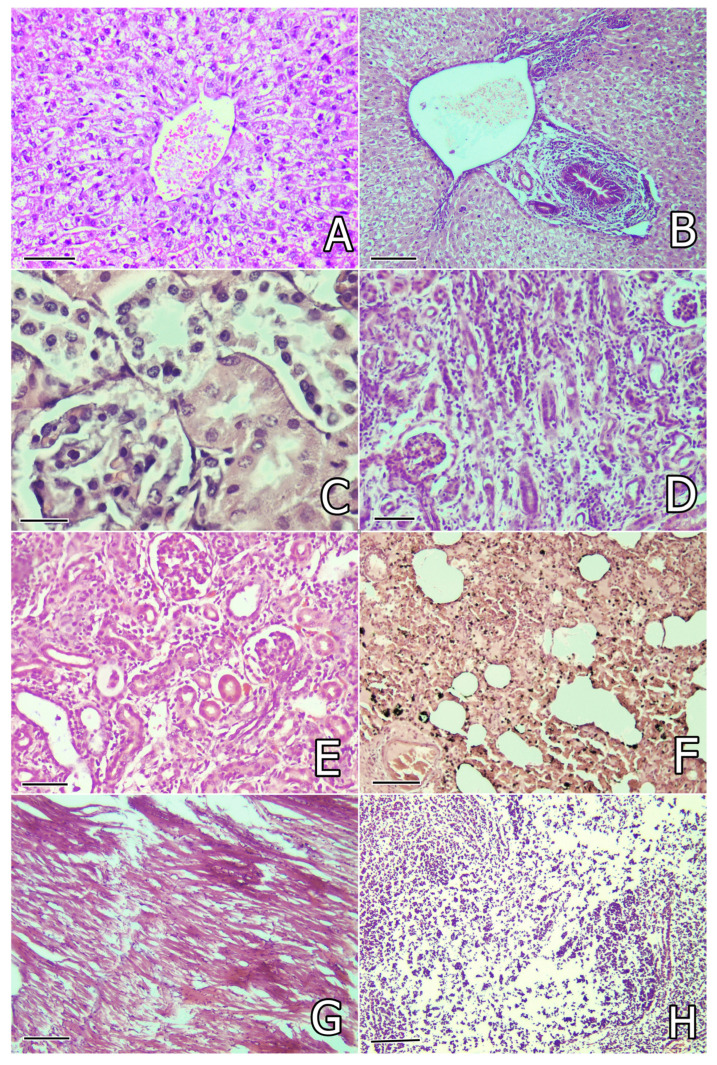
(**A**) Granular degeneration in hepatocytes and activation of Kupfer cells in the liver of a rabbit from the OTA group on the 30th day of the study. Liver. H/E (hematoxylin and eosin) ×240, bar = 42 µm; (**B**) bile duct proliferation, peribiliary fibrosis, and edema in the liver of a rabbit from the OTA group on the 80th day of the study. Liver. H/E ×200, bar = 50 µm; (**C**) granular degeneration and epithelial desquamation in the proximal tubules of the kidneys in a rabbit from the OTA group on the 30th day of the study. Kidney. H/E ×300, bar = 33 µm; (**D**) mononuclear and connective tissue proliferation, atrophy of some tubules and sclerosis of glomeruli in the kidneys in a rabbit from the OTA group on the 80th day of the study. Kidney. H/E ×240, bar = 42 µm; (**E**) mild mononuclear and connective tissue proliferation in the interstitium and atrophy of some tubules in the kidneys in a rabbit from the WS + OTA group on the 80th day of the study. Kidney. H/E ×240, bar = 42 µm; (**F**) focal catarrhal pneumonia, hyperemia, and many siderocytes in the lung in a rabbit from the OTA group on the 80th day of the study. Lung. H/E ×240, bar = 42 µm; (**G**) degenerative and lytic lesions or irregular staining of myofibrils (eosinophilia) in the heart of a rabbit from the OTA group on the 80th day of the study. Heart. H/E ×200, bar = 50 µm; (**H**) degenerative lesions and depletion of white pulp cells in the spleen of a rabbit from the OTA group on the 30th day of the study. Spleen. H/E ×200, bar = 50 µm.

**Figure 3 toxins-17-00507-f003:**
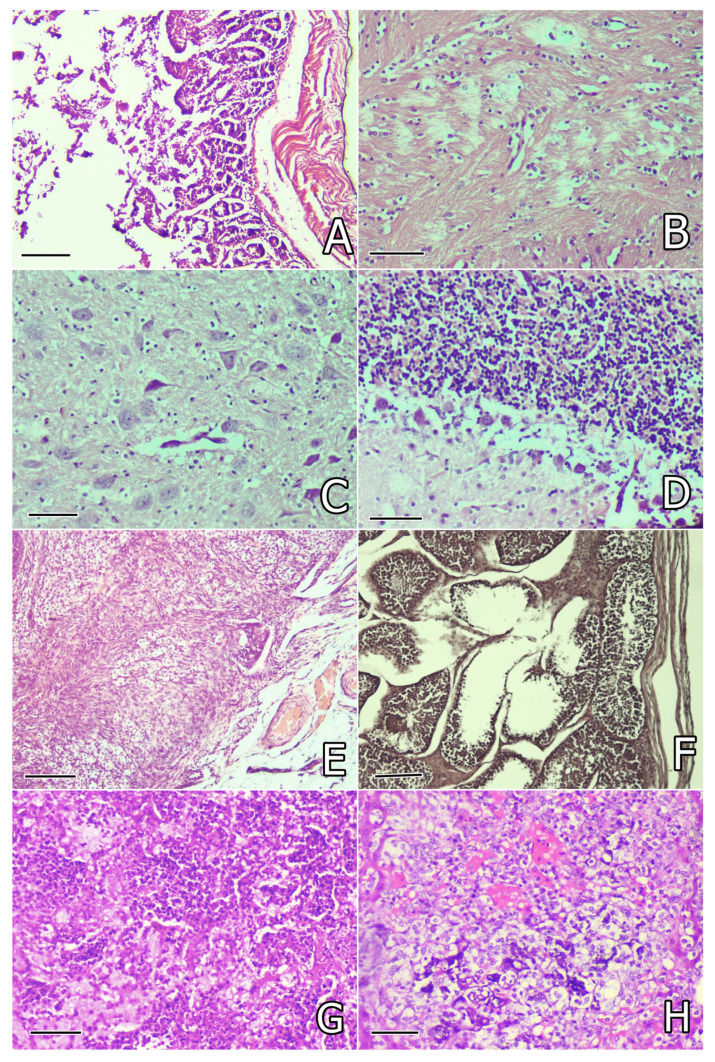
(**A**) Degenerative lesions in the superficial and glandular epithelium of the intestinal mucosa of a rabbit from the OTA group on the 30th day of the study. Small intestine. H/E ×200, bar = 50 µm; (**B**) lytic lesions and pericapillary or pericellular oedema in the brain substance of a rabbit from the OTA group on the 30th day of the study. Brain. H/E ×240, bar = 42 µm; (**C**) lysis or pyknosis in the tigroid substance in neurons (tigrolysis or tigropyknosis) in the brain of a rabbit from the OTA group on the 30th day of the study. Brain. H/E ×240, bar = 42 µm; (**D**) Edematous and lytic lesions in the region of Purkinje cells in the cerebellum of a rabbit from the OTA group on the 30th day of the study. Cerebellum. H/E ×240, bar = 42 µm; (**E**) congestion of blood vessels, edema, and connective tissue proliferation in the ovary of a rabbit from the OTA group on the 30th day of the study. Ovary. H/E ×200, bar = 50 µm; (**F**) mild degenerative lesions in part of the seminiferous tubules in the testicle of a rabbit from the OTA group on the 80th day of the study. Testicle. H/E ×200, bar = 50 µm; (**G**) croupous pneumonia with fibrin accumulation in the alveoli and interstitial tissue in the lung of a rabbit from the OTA group that died of pasteurellosis. Lung. H/E ×240, bar = 42 µm; (**H**) severe focal degeneration and necrotic changes in hepatocytes and fibrin accumulation in capillaries. Liver. H/E ×240, bar = 42 µm.

**Table 1 toxins-17-00507-t001:** Mean values of body weight (b.w.) of rabbits supplemented with various experimental additives and/or OTA for 80 days (n = 8) *.

Group	Body Weight (b.w.)
	(kg)
OTA	2.14 ± 0.05 ^a^
WS + OTA	2.7 ± 0.04 ^ab^
CA + OTA	2.87 ± 0.04 ^ab^
SM + OTA	2.69 ± 0.03 ^ab^
SIL + OTA	2.71 ± 0.03 ^ab^
Control	3.19 ± 0.04 ^b^

± SEM (standard error of the mean). Means sharing the same superscript are not significantly different from each other (*p* < 0.05). ^a^ Indicates significant difference compared to the control group (*p* < 0.05). ^b^ Indicates significant difference compared to the OTA group (*p* < 0.05). * The number of rabbits studied was 6 for the OTA group, 7 for the CA + OTA and SM + OTA groups, and 8 for the WS + OTA, SIL + OTA and control groups due to the death of a few rabbits from pasteurellosis.

**Table 2 toxins-17-00507-t002:** Mean values of antibody titer in groups of rabbits (n = 8) * given various putative antidotes and/or OTA on day 28 after vaccination against RHDV ^#^.

Group	0 Day	28th Day
OTA	0.10 ± 0.021	0.51 ± 0.06 ^a^
WS + OTA	0.09 ± 0.001	0.93 ± 0.20 ^bc^
CA + OTA	0.09 ± 0.005	0.53 ± 0.07 ^a^
SM + OTA	0.10 ± 0.008	0.60 ± 0.06 ^a^
SIL + OTA	0.09 ± 0.002	1.14 ± 0.12 ^bc^
Control	0.10 ± 0.011	1.01 ± 0.02 ^bc^

± SEM (standard error of the mean). **^#^** All rabbits were immunized at the age of 37 days (immediately after beginning of the experiment) against RHDV. The blood for serological examinations was taken at the age of 65 days (28 after immunization). ^a^ Indicates significant difference compared to the control group (*p* < 0.05). ^b^ Indicates significant difference compared to the OTA group (*p* < 0.05). ^c^ Samples with an OD value (at a dilution of 1/200) above 0.9 indicate sufficiently developed immune protection against RHDV infection. * The number of rabbits studied was 6 for the OTA group, 7 for the CA + OTA and SM + OTA groups, and 8 for the WS + OTA, SIL + OTA and control groups due to the death of a few rabbits from pasteurellosis.

**Table 3 toxins-17-00507-t003:** Pathomorphological changes in the tissues of the internal organs of rabbits from different experimental groups on day 30 and day 80 of the experiment and rabbits that died on days 12–14.

Pathomorphological Changes	OTA + SIL	OTA + SM	OTA + WS	OTA + CA	OTA
**Liver**					
Degenerative lesions in hepatocytes	++	++	++	++	++++
Congestion or perivascular mononuclear infiltration	+	+	+	+	+++
Activation of endothelial cells and Kupffer’s cells	+	+	+	+	++
Bile duct proliferation and fibrosis or edema around bile ducts	++	++	+	+	+++
Focal necroses and/or fibrin accumulation in rabbits with pasteurellosis #	-	++	-	++	+++++
**Kidneys**					
Granular degeneration in proximal tubules	++	++	++	++	++++
Congestion of peritubular capillaries	+	+	+	+	++
Inflammatory cells infiltration in interstitium	+	+	+	+	+++
Endothelial proliferation in peritubular capillary	+	+	+	+	++
Proliferation of connective tissue on the 80th day *	+	-	+	-	+++
Sclerosis of some glomerules on the 80th day *	-	+	+	+	+++
Tubular atrophy and retention cysts on the 80th day *	-	-	+	-	+++
Thrombosis of some vessels (fibrin clots) in rabbits with pasteurellosis #	-	+	-	+	++
**Lung**					
Congestion of vessels and presence of siderocytes	+	+	+	+	++
Peribronchial or perivascular mononuclear infiltration	+	+	+	+	++
Focal catarrhal pneumonia	+	+	+	+	++
Purulent or croupous pneumonia and/or abscesses in rabbits with pasteurellosis #	-	+++++	-	+++++	+++++
Fibrinous pleurisy and pericarditis, incl. adhesions in rabbits with pasteurellosis #	-	+++	-	+++	+++++
Thrombosis (fibrin clots) in vessels in rabbits with pasteurellosis #	-	+++	-	+++	+++++
**Myocardium**					
Vascular congestion	+	+	+	+	++
Mononuclear cells infiltration	+	+	-	+	++
Slight granular degeneration and/or lytic changes	+	+	+	+	++
Strong granular degeneration and/or thrombosis (fibrin clots) in vessels in rabbits with pasteurellosis #	-	++	-	++	+++++
**Spleen**					
Degenerative lesions in white pulp	-	+	-	+	++
Depletion of cells in white pulp	-	+	+	++	+++
White pulp reduction and predomination of red pulp in rabbits with pasteurellosis #	-	++	-	++	+++
**Thymus**					
Degenerative lesions or depletion of cells in cortex	+	+	+	++	+++
**Intestines**					
Degenerative lesions of surface/glandular epithelium	+	+	+	+	++
Vascular congestion	+	+	+	+	++
**Brain**					
Lytic/pyknotic changes in tigroid substance of neurons	+	+	+	+	++
Lytic changes in brain substance	+	+	+	+	++
Pericapillary and pericellular edema	+	+	+	+	++
Pericapillary infiltration with lymphocytes	-	-	-	-	++
Focal proliferation of microglia	-	-	-	-	++
**Cerebellum**					
Lytic changes in *lamina medullaris* or Purkinje cells	+	+	+	+	++
**Ovary**					
Connective tissue proliferation and interstitial edema on the 80th day *	+	+	+	+	++
Vascular congestion	-	-	-	-	+
**Testicles**					
Degenerative lesions in seminiferous tubules on the 80th day *	+	+	+	+	++
Interstitial edema on the 80th day *	+	+	+	+	++

+ Minor lesions or lesions found occasionally in a few rabbits. ++ Minor lesions in all rabbits or moderate lesions in less than half of the rabbits. +++ Moderate lesions in all rabbits and/or severe lesions in less than half of the rabbits. ++++ Strong lesions in more than half of the rabbits, but not all of them. +++++ Severe lesions in all rabbits. * Lesions observed in rabbits on the 80th day. # Lesions observed in rabbits that died from pasteurellosis. SIL: Silymarin; SM: *Silybum marianum;* WS: *Withania somnifera;* CA: *Centella asiatica;* OTA: *ochratoxin A*.

**Table 4 toxins-17-00507-t004:** Concentrations of ochratoxin A (OTA) and supplements in standard complete feeds of rabbits.

Group	OTA in Feed	Herbal Supplements in Feed
	(ppm-mg/kg)	(ppm or mg/kg.b.w.)
OTA	2	none
WS + OTA	2	4000 ppm *W. somnifera radix* powder (around 200 mg/kg b.w.)
CA + OTA	2	4600 ppm *C. asiatica leaves* powder (around 230 mg/kg b.w.)
SM + OTA	2	5000 ppm *S. marianum fruits* powder (around 250 mg/kg b.w.)
SIL + OTA	2	25,000 ppm Extract Powder of Silymarin (around 1250 mg/kg b.w.)
CONTROL	none	none

## Data Availability

The original contributions presented in this study are included in the article. Further inquiries can be directed to the corresponding author(s).

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
