# Peer review of "Susceptibility to Secondary Bacterial Infections in Growing Rabbits Exposed to Ochratoxin A and Protected or Not by Herbal Supplements"

_toxins, 2025, doi:10.3390/toxins17100507_

Round 1

Reviewer 1 Report

Comments and Suggestions for Authors

In the chapter on Materials and Methods, there is very little data on obtaining the ochratoxin used and also on the chromatographic method; there are only two bibliographical references mentioned from previous studies, but since the mycotoxin was introduced into the feed, more data should have been provided on the recovery rate, LOD, and chromatograms, which could even have been included in supplementary material. Similarly, the structure and composition of the standard rabbit feed was only mentioned without any data on its content. An update to the bibliographic references is required, considering that 70% of the articles cited are more than 10 years old. 

Author Response

A detailed answer to Referee 1

  1. In the chapter on Materials and Methods, there is very little data on obtaining the ochratoxin used and also on the chromatographic method; there are only two bibliographical references mentioned from previous studies, but since the mycotoxin was introduced into the feed, more data should have been provided on the recovery rate, LOD, and chromatograms, which could even have been included in supplementary material.

Answer: OTA was produced in Imperial College of Science, Technology and Medicine in London under my previous fellowships (PIRSES-GA-2012-316067 HERBAL PROTECTION and UK Royal Society Joint Project with Central and Eastern Europe) as described previously in the paper (Harris and Mantle, 2001). This paper is now added into the Reference as follow:

  1. Harris J.P.; Mantle, P.G. The biosynthesis of ochratoxins by Aspergillus ochraceus. Phytochemistry 2001; 58, 709–716.

The final OTA-content in the feed was also studied by HPLC with detection limit 0,15 µg/kg (ppb) and average recovery of 80% as previously described [44]. Some new information and a new reference is now added in this regard:

  1. Stoev S.D.; Dutton, M.; Njobeh, P.; Mosonik, M.; Steenkamp, P. Mycotoxic nephropathy in Bulgarian pigs and chickens: complex aetiology and similarity to Balkan Enedemic Nephropathy. Food Addit. Contam. Part A. 2010, 27(1), 72-88, http://www.tandfonline.com/doi/pdf/10.1080/02652030903207227

  1. Similarly, the structure and composition of the standard rabbit feed was only mentioned without any data on its content.

Answer: Now the structure and composition of the standard rabbit feed was given as referee suggested as follow: “Feed composition: alfalfa meal, barley, oats, wheat bran, sunflower meal, soybean meal, wheat, corn, sugar beet granules and some minerals, incl.: calcium carbonate, monocalcium phosphate, L lysine, DL methionine, sodium chloride, lecithin, vitamin-microelement premix, L threonine, sodium bicarbonate, rice bran, choline chloride, sunflower oil, acidifier.

Analytical composition of feed: crude protein (17%), crude fat (3.6%), crude fiber (15%), crude ash (7.3%), calcium (0.91%), phosphorus (0.52%), sodium (0.27%).

Additives in 1 kg of feed: vitamin A – 6000 IU/kg, vitamin D3 – 600 IU/kg, vitamin E - 24.58 IU/kg; trace elements (mg/kg): ferrous sulfate monohydrate (106.6), copper sulfate pentahydrate (18.5), manganese oxide (72.6), zinc oxide (96.5), calcium iodate (1.06), selenium (0.45); antioxidants (mg/kg): E321 (4.0), E320 (1.5), E330 (2.0).”

  1. An update to the bibliographic references is required, considering that 70% of the articles cited are more than 10 years old.

Answer: This suggestion by the Referee is now followed and some new references are now added at the place of some old references. Unfortunately, it is not possible to quote only new references, because of some critical points in subsequent discussion of the same references as well as due to the circumstance that there are only scarce data (not so fresh) about OTA-intoxication in rabbits and possible natural protective measures.

Some new references are now added:

[26]. Vajdi, M.; Adeli, S.; Karimi, A.; Asghariazar, V.; Jazani, A.M.; Azgomi, R.N. The Impact of Silymarin on Inflammation and Oxidative Stress: A Systematic Review and Meta-Analysis of Randomized Controlled Trials. Int. J. Clin. Pract. 2025, 1, 3985207, https://doi.org/10.1155/ijcp/3985207

[27]. Mohammadi, S.; Ashtary-Larky, D.; Asbaghi, O.; Farrokhi, V.; Jadidi, Y.; Mofidi, F.; Mohammadian, M.; Afrisham, R. Effects of silymarin supplementation on liver and kidney functions: A systematic review and dose–response meta‐analysis. Phytotherapy Research 2024, 38(5), 2572-2593.

[32]. Seo, M.G.; Jo, M.J.; Hong, N.I.; Kim, M.J.; Shim, K.S.; Shin, E.; Lee, J.J.; Park, S.J. Anti-inflammatory and anti-vascular leakage effects by combination of Centella asiatica and Vitis vinifera L. leaf extracts. Evid Based Complement Alternat Med. 2021, 1, 7381620, https://doi.org/10.1155/2021/7381620.

[39] Mikulska, P.; Malinowska, M.; Ignacyk,M.; Szustowski, P.; Nowak, J.; Pesta, K.; Szeląg, M.; Szklanny, D.; Judasz, E.; Kaczmarek, G.; Ejiohuo, O.P.; Paczkowska-Walendowska, M.; Gościniak, A.; Cielecka-Piontek, J. Ashwagandha (Withania somnifera)—Current research on the health-promoting activities: A Narrative Review. Pharmaceutics 2023, 15(4), 1057, doi: 10.3390/pharmaceutics15041057

Reviewer 2 Report

Comments and Suggestions for Authors

This study aims to elucidate possible protective effects of herbal supplements including C. asiatica, W. somnifera, S. marianum and Silymarin applied to rabbit diet on the toxic effects of OTA. This is an interesting piece of work and should be of interest in the field. However, the following issues need to be addressed for a possible publication in Toxins:

Abstract

  1. Line 7-14: this sentence is too long and should be broken down into shorter sentences for clarity and readability.
  2. Line 12-16: rephrase “a potent OTA-induced immunosuppression on humoral immune response was seen” for clarity.
  3. Some key quantitative data should be provided to avoid vague presentation like “appeared to have protective effects” in line 22-23 and other places in abstract.

Introduction

  1. There are several grammatical errors which needs to be double-checked.
  2. Some very old references should be removed and instead new citation in the last three years should be included.

Materials and methods

  1. A reference citation each should be provided for the procedures detailed in this section.
  2. In line 111, sterilization temperature and time that doe not degrade OTA should be specified.
  3. In line 117, the HPLC method should be briefly described and the detection limit should be specified.
  4. Line 127-131: dose calculation details are insufficient and the actual mg/kg body weight equivalents should be added.
  5. The study design in section 2.3 should be represented figuratively for easy readability.
  6. Line 189-190: The sedative brand/source should be identified.
  7. Line 192-210: the positive and negative control sera should be indicated for ELISA.
  8. In section 2.9, the statistical software used should be specified along with version and purchase details (city and country) should be specified.

Results

  1. Line 240-253: incidence percentages should be added.
  2. Table 2: the sample size should be added in caption.
  3. Table 3: the authors should provide p-values for groups comparisons.
  4. Figures 1-3: the scale bars and staining methods should be included in the corresponding figure legend/captions.

Discussion

  1. Line 486-499: dose-response limitation (single OTA level only) should be mentioned.
  2. An explicit statement on small group size (n=8) and potential sex differences should be included.

References

  1. Double-check if all the references in the reference list are cited in the text.
  2. Double-check if all the references conform to journal’s referencing style and consistent with the usage of DOIs, italics and punctuation.

General

  1. All the abbreviations in the entire manuscript should be defined in full at the first instance and abbreviated thereafter.
  2. Numerous typographical errors needs to be corrected throughout the manuscript.

Author Response

A detailed answer to Referee 2

Abstract

  1. Line 7-14: this sentence is too long and should be broken down into shorter sentences for clarity and readability.

Answer: This sentence is now separated to three different sentences for more clarity and readability as follow: “Protective effects of herbal feed supplements Silybum marianum, Silymarin, Withania somnifera and Centella asiatica against ochratoxin A (OTA) toxicity were studied in 48 New-Zealand-White rabbits (37-day-old) during 80-day experiment. OTA was given at 2 ppm, whereas Silybum marianum, Silymarin, Withania somnifera and Centella asiatica were given at feed levels 5000 ppm, 25000 ppm, 4000 ppm and 4600 ppm respectively. All rabbits were immunized against Rabbit-Hemorrhagic-Disease-Virus (RHDV).

  1. Line 12-16: rephrase “a potent OTA-induced immunosuppression on humoral immune response was seen” for clarity.

Answer: This sentence is now changed to: “OTA was found to induce immunosuppressive effect on humoral immune response”, in other to improve clarity as Referee suggested.

  1. Some key quantitative data should be provided to avoid vague presentation like “appeared to have protective effects” in line 22-23 and other places in abstract.

Answer: This sentence is now specified to: “All investigated herbal supplements appeared to have stronger protective effects against OTA-induced damages on kidneys and liver, but slighter protective effects towards lung, myocardium, spleen, brain, intestine, testicles and ovary”. Unfortunately, it is not possible to give more specified information in the Abstract, due to the given limitation of words.

Introduction

  1. There are several grammatical errors which needs to be double-checked.

Answer: The entire manuscript is now checked for possible grammatical errors and clarity of sentences and necessary corrections were made.

  1. Some very old references should be removed and instead new citation in the last three years should be included.

Answer: This suggestion by the Referee is now followed and some new references are now added at the place of some old references. Unfortunately, it is not possible to quote only new references, because of some critical points in subsequent discussion of the same references as well as due to the circumstance that there are only scarce data (not so fresh) about OTA-intoxication in rabbits and possible natural protective measures.

Some new references are now added:

[26]. Vajdi, M.; Adeli, S.; Karimi, A.; Asghariazar, V.; Jazani, A.M.; Azgomi, R.N. The Impact of Silymarin on Inflammation and Oxidative Stress: A Systematic Review and Meta-Analysis of Randomized Controlled Trials. Int. J. Clin. Pract. 2025, 1, 3985207, https://doi.org/10.1155/ijcp/3985207

[27]. Mohammadi, S.; Ashtary-Larky, D.; Asbaghi, O.; Farrokhi, V.; Jadidi, Y.; Mofidi, F.; Mohammadian, M.; Afrisham, R. Effects of silymarin supplementation on liver and kidney functions: A systematic review and dose–response meta‐analysis. Phytotherapy Research 2024, 38(5), 2572-2593.

[32]. Seo, M.G.; Jo, M.J.; Hong, N.I.; Kim, M.J.; Shim, K.S.; Shin, E.; Lee, J.J.; Park, S.J. Anti-inflammatory and anti-vascular leakage effects by combination of Centella asiatica and Vitis vinifera L. leaf extracts. Evid Based Complement Alternat Med. 2021, 1, 7381620, https://doi.org/10.1155/2021/7381620.

[39] Mikulska, P.; Malinowska, M.; Ignacyk,M.; Szustowski, P.; Nowak, J.; Pesta, K.; Szeląg, M.; Szklanny, D.; Judasz, E.; Kaczmarek, G.; Ejiohuo, O.P.; Paczkowska-Walendowska, M.; Gościniak, A.; Cielecka-Piontek, J. Ashwagandha (Withania somnifera)—Current research on the health-promoting activities: A Narrative Review. Pharmaceutics 2023, 15(4), 1057, doi: 10.3390/pharmaceutics15041057

Materials and methods

  1. A reference citation each should be provided for the procedures detailed in this section.

Answer: It is now done accordingly.

  1. In line 111, sterilization temperature and time that doe not degrade OTA should be specified.

Answer: This is now done as follow: “…..then sterilised at 80°C for 1 hour. OTA was produced in Imperial College of Science, Technology and Medicine in London under my previous fellowships (PIRSES-GA-2012-316067 HERBAL PROTECTION and UK Royal Society Joint Project with Central and Eastern Europe) as described previously in the paper (Harris and Mantle, 2001). This paper is now added into the Reference as follow:

  1. Harris J.P.; Mantle, P.G. The biosynthesis of ochratoxins by Aspergillus ochraceus. Phytochemistry 2001; 58, 709–716.

  1. In line 117, the HPLC method should be briefly described and the detection limit should be specified.

Answer: The final OTA-content in the feed was also studied by HPLC with detection limit 0,15 µg/kg (ppb) and average recovery of 80% as previously described [44]. Some new information and a new reference is now added in this regard:

  1. Stoev S.D.; Dutton, M.; Njobeh, P.; Mosonik, M.; Steenkamp, P. Mycotoxic nephropathy in Bulgarian pigs and chickens: com-plex aetiology and similarity to Balkan Enedemic Nephropathy. Food Addit. Contam. Part A. 2010, 27(1), 72-88, http://www.tandfonline.com/doi/pdf/10.1080/02652030903207227

  1. Line 127-131: dose calculation details are insufficient and the actual mg/kg body weight equivalents should be added.

Answer: There are many factors, which have been taken into account when calculating the doses of the herbal supplements. The same were calculated having into account the dose conversion factors available in the literature and body weight of the rabbits, the WHO norms and other toxicological guidelines as described previously [38], in order to be near to the effective doses against OTA-toxicity. The available data from experimental studies were also taken into consideration [20,28,35,41]. Some additional adjustments were made based on the rabbit breed and age.

  1. Stoev, S.D. Biocontrol Agents and Natural Feed Supplements as a Safe and Cost-Effective Way for Preventing Health Ailments Provoked by Mycotoxins. Foods 2025, 14(11), 1960, https://www.mdpi.com/2304-8158/14/11/1960

The actual mg/kg body weight equivalents are now added in Table 1 as Referee suggested.

Unfortunately, the available data in the literature mainly address the herbal extracts, but not the herbal powder, which makes the direct comparison of the doses used in various experiments almost impossible.

Some additional changes in the given information were also made in other make it more clear.

In order to justify the final dose in the diet we also visited some local Open Herbs Market in Johannesburg (under previous Marie Curie fellowship “HERBAL PROTECTION”) for collecting more information from the source about the healing or protective effects of some local herbs and their doses used by traditional healers, and also made a direct contact and meetings with the leading herbal expert in South Africa and India, e.g.: 

  1. a) Prof. Ben-Eric van Wyk (Chairman of the Aloe Council in SA, Chairman of the Indigenous Plant Use Forum in SA, Member of the Association for African Medicinal Plant Standards – AAMPS, Member of the Presidential Task Team on African Traditional Medicine) from Dept of Botany, Fac of Science niversity of Johannesburg
  2. b) Prof. J. van Staden, Director of Research Centre for Plant Growth and Development, School of Life Science, University of KwaZulu-Natal, Pietermaritzburg, South Africa
  3. c) Prof. JN (Kobus) Eloff, Leader Phytomedicine Programme, Department of Paraclinical Sciences, Faculty of Veterinary Science, University of Pretoria, South Africa –
  4. d) Dr Rajesh Arora, Institute of Nuclear Medicine and Allied Sciences, Defence Research and Development Organization, India

All these leading scientists in the field of herbal protection help us to make some additional adjustment about the dose of the herbs in rabbits having in mind the absence of suitable information in the literature in this regard and we are very thankful for this help. 

  1. The study design in section 2.3 should be represented figuratively for easy readability.

Answer: Unfortunately, I cannot represent the study design figuratively. I found that such a representation is almost impossible or at least enormously difficult for me. I think, that the changes made in Table will facilitate the readability to some extent.

  1. Line 189-190: The sedative brand/source should be identified.

Answer: This information is now added according to referee suggestion as follow: “1 ml blood was taken from v. auricularis (or v. jugularis) of each rabbit, after pre-sedation with Acepromazine (Netherland, 3440 AB Woerden) at dose 0.75 mg/kg i.m.”

  1. Line 192-210: the positive and negative control sera should be indicated for ELISA.

Answer: This information is now added according to referee suggestion as follow: “100 µL of the diluted samples (positive and negative control serum) were added to the wells of the plate……”.

“The validation of the assay is performed when OD Positive control / OD Negative control > 10. Cut off calculation: Cut off positive/Negative is 0.300. Samples with an OD higher than 0.300 must be considered as positives and samples with OD lower than 0.300 must be considered as negatives. The sera titter is the highest dilution with an OD higher than 0.300.”

  1. In section 2.9, the statistical software used should be specified along with version and purchase details (city and country) should be specified.

Answer: The statistical software used along with the version (city and town) are now specified as referee suggested as follow:  “This was done by using the statistical software GraphPad InStat version 3.10 (San Diego, USA).”

Results

  1. Line 240-253: incidence percentages should be added.

Answer: It is now added according to referee suggestions as follow: “In addition, 4 rabbits from OTA group (50%) and a few rabbits from the groups supplemented with C. asiatica (CA+OTA) (2 rabbits – 25%), S. marianum (SM+OTA) (1 rabbit – 12,5%) and W. somnifera (WS+OTA) (1 rabbit – 12,5%) showed difficulty breathing and runny nose or snuffles seen at the end of the second week of the experiment, and some of them died within several days as follow: 2 rabbits (1 male and 1 female – 25%) from OTA group died on day 12 from the beginning of experiment, 1 female rabbit from CA+OTA group (12,5%) died on day 13 and 1 female rabbit from SM+OTA group (12,5%) died on day 14 from the beginning of experiment.”

  1. Table 2: the sample size should be added in caption.

Answer: The sample size is now added in caption using an “asterisk -*” according to referee suggestions and some explanations are given in the bottom of the table as follow: “* - the number of rabbits studied was 6 for OTA group, 7 for CA+OTA and SM+OTA groups and 8 for WS+OTA, SIL+OTA and CONTROL groups due to the death of a few rabbits from pasteurellosis”

  1. Table 3: the authors should provide p-values for groups comparisons.

Answer: The p-values are now added as Referee suggested.

  1. Figures 1-3: the scale bars and staining methods should be included in the corresponding figure legend/captions.

Answer: The staining methods is given in abbreviations H/E (Hematoxylin and Eosin). Now this abbreviation is defined in full at the first instance and abbreviated thereafter as referee suggested. In regard to the “scale bars”, such information is usually given, when the magnification is not mentioned. In all histological pictures, such magnifications are given ×240, ×300, etc. In my experience as a reviewer and editorial board member in some journals (including “Toxins”), I have never asked for such additional information, and such scale bars were not given (in addition to magnifications) in my previous papers in “Toxins”. Nevertheless, such additional information, such as “scale bars” is now included in this paper as referee suggested.

Discussion

  1. Line 486-499: dose-response limitation (single OTA level only) should be mentioned.

Answer: Thanks a lot for this suggestion. Now the following text is added to comply with this recommendation as follow: “Due to ethical considerations and limitations in the number of animals used, as well as due to the dose-response limitation (only one feed level of OTA was studied) and sex differences inside the groups, the observed statistical fluctuations between the groups were not always sufficient to establish confidential differences between them.”

  1. An explicit statement on small group size (n=8) and potential sex differences should be included.

Answer: Thanks a lot for this suggestion. Now the following text is added to comply with this recommendation as follow: “Due to ethical considerations and limitations in the number of animals used, as well as due to the dose-response limitation (only one feed level of OTA was studied) and sex differences inside the groups, the observed statistical fluctuations between the groups were not always sufficient to establish confidential differences between them.”

References

  1. Double-check if all the references in the reference list are cited in the text.

Answer: It is now done accordingly

  1. Double-check if all the references conform to journal’s referencing style and consistent with the usage of DOIs, italics and punctuation.

Answer: It is now done accordingly

General

  1. All the abbreviations in the entire manuscript should be defined in full at the first instance and abbreviated thereafter.

Answer: It is now done accordingly

  1. Numerous typographical errors needs to be corrected throughout the manuscript.

Answer: The entire manuscript is now checked for possible typographical errors and clarity of sentences and necessary corrections were made.

Round 2

Reviewer 1 Report

Comments and Suggestions for Authors

The authors partially met the requirements regarding the bibliography (where they still have work to do) and the feed recipe used for rabbits, but to a very small extent regarding the chromatographic method, where they provided few data. 

Author Response

Comment: The authors partially met the requirements regarding the bibliography (where they still have work to do) and the feed recipe used for rabbits, but to a very small extent regarding the chromatographic method, where they provided few data.

Answer: Dear colleague, as I explained before, OTA was produced in Imperial College of Science, Technology and Medicine in London under my previous fellowships (PIRSES-GA-2012-316067 HERBAL PROTECTION and UK Royal Society Joint Project with Central and Eastern Europe) as described previously in the paper (Harris and Mantle, 2001) and this paper was added into the Reference:

  1. Harris J.P.; Mantle, P.G. The biosynthesis of ochratoxins by Aspergillus ochraceus. Phytochemistry 2001; 58, 709–716.

In my experience as a reviewer and editorial board member of some journals (including “Toxins”), I have never asked for such additional information that has been previously described in other papers. Therefore, it has been provided briefly by citing the relevant papers. Moreover, the purpose of this experiment, was to elucidate possible protective effects of herbal supplements applied to the rabbit diet towards the toxic effects of OTA, but not toxicological investigations of animal feeds. This was the reason to avoid such excessive data in the Material and methods section. Nevertheless, such additional data on chromatographic methods are now included in this paper, as suggested by the reviewer as follow:

“A multi-mycotoxin extraction method (multi-mycotoxin screen was used to analyze the feed samples [44]. In the same method, two extracts can be generated in one step: a neutral fraction and an acid fraction (containing OTA and some other mycotoxins).

A two dimensional thin layer chromatographic (TLC) technique was used initially as 20 µl of the acid fractions (containing OTA) dissolved in dichloromethane (DCM) were spotted on silica gel TLC plates (about 1 cm from the edge of a silica gel TLC plate) and dried in a warm stream of air. The spotted plates were then developed in TLC tanks using two solvents Diethyl Phthalate (DEP) and Toluene-Ethyl Acetate-Formic Acid (TEF) in two-dimensional directions and were dried after each development. The plates were visualized under ultraviolet (UV) light at 366 nm for the presence of any fluorescent and were compared with the standard plate for OTA. The blue fluorescing spot with retardation factors RF1 (50) and RF2 (30) for OTA was then compared with the RF values of a standard for OTA [44].

HPLC analysis and quantification of OTA was performed using Shimadzu system (Kyoto, Japan) with fluorescence detector (RF-10AXL) and Symmetry column (250 ×4.6 mm internal diameter) as previously described [44]. The procedure included the following parameters: excitation & emission wavelength (333 & 477 nm), mobile phase—acetonitrile-water (50:50, v/v), flow rate—0.8 ml/min, injection volume of analyte & standard — 10 μl per sample, running time—30 min, column temperature— 30°C [44]. Limit of detection (LOD) was about 0,03 µg/kg (ppb) and average recovery was above 80%. The identification of the mycotoxin was based on the comparison of the UV spectra and the retention times of the detected peaks with those of the standard. OTA was quantified using peak area and external standard calibration. Selected feed samples (free of mycotoxins) were spiked with known levels of OTA standard and were processed in a similar manner as the other samples in order to establish recovery rate for OTA.”

Reviewer 2 Report

Comments and Suggestions for Authors

The authors have satisfactorily addressed all the comments raised by reviewers and substantially improved the overall quality of the article. Therefore, I recommend accepting this article for publication in Toxins.

Author Response

Thanks a lot for your efforts for improving the paper and for your final decision.